# Preparation and Thermomechanical Properties of Ketone Mesogenic Liquid Crystalline Epoxy Resin Composites with Functionalized Boron Nitride

**DOI:** 10.3390/polym12091913

**Published:** 2020-08-25

**Authors:** Yi-Sheng Lin, Steve Lien-Chung Hsu, Tsung-Han Ho, Li-Cheng Jheng, Yu-Hsiang Hsiao

**Affiliations:** 1Department of Materials Science & Engineering, National Cheng Kung University, Tainan 701-01, Taiwan; sheng-013@hotmail.com; 2Product Characterization, Advanced Semiconductor Engineering, Inc., Kaohsiung 801-70, Taiwan; Hunter_Hsiao@aseglobal.com; 3Department of Chemical & Materials Engineering, National Kaohsiung University of Science and Technology, Kaohsiung 807-78, Taiwan; thho@nkust.edu.tw (T.-H.H.); lcjheng@nkust.edu.tw (L.-C.J.)

**Keywords:** liquid crystalline epoxy resin, boron nitride, composites, surface modification, thermomechanical properties

## Abstract

In order to enhance the thermomechanical behaviors of epoxy molding compounds, the hexagonal boron nitride (h-BN) fillers were incorporated in a ketone mesogenic liquid crystalline epoxy (K–LCE) matrix to prepare a high-performance epoxy composites. The h-BN was modified by surface coupling agent 3-aminopropyltriethoxysilane (APTES). The grafting of silane molecules onto the surface of BN fillers improved the compatibility and homogeneous dispersion state of BN fillers in the K–LCE matrix with a strong interface interaction. The surface-modified BN fillers were characterized using Fourier transform infrared spectroscopy. The thermomechanical properties and morphologies of K–LCE/BN composites loading with different contents of modified BN fillers, ranging from 0.50 to 5.00 wt%, were investigated. These results show that modified BN fillers uniformly dispersed in K–LCE matrix, contributing to the enhancement in storage modulus, glass transition temperatures, impact strength and reduction in the coefficient of thermal expansion (CTE). The thermal stability and char yield of the K–LCE/BN composites were increased by increasing the amount of modified BN fillers and the thermal decomposition temperatures of composites were over 370 °C. The thermal conductivity of the K–LCE/BN composites was up to 0.6 W/m·K, for LC epoxy filled with 5.00-wt%-modified BN fillers. Furthermore, the K–LCE/BN composites have excellent thermal and mechanical properties compared to those of the DGEBA/BN composites.

## 1. Introduction

With the development of electronic packaging technology, the heat management is becoming increasingly important, which has led to the increased demand for higher density and higher power integrated circuits (ICs). When high-power–density devices are worked, a large amount of heat is produced. The issue of heat dissipation must be improved to prevent overheating [1,2,3].

Epoxy molding compounds (EMCs) and underfills has often been used as electrical encapsulating materials for microelectronic devices to provide protection of the IC devices [4]. However, common epoxy resins have poor thermal properties such as high CTE and low thermal conductivity (0.17–0.20 W/m·K), but these properties of epoxides can be improved by the rigid rod structures with liquid crystalline characteristics introduced [5,6,7,8]. We have reported that the liquid crystalline (LC) epoxy resin exhibited much lower CTE and a higher thermal conductivity (0.34 W/m·K) compared to those of a traditional epoxy resin (DGEBA) [9]. In general, the ICs are encapsulated by silica-filled epoxy composites, but the thermal conductivity of the composites is less than 1.0 W/m·K, which is far below the demand of ICs heat dissipation [10]. It can be enhanced by incorporation of inorganic fillers with a high thermal conductivity and electrical insulation, such as alumina (Al_2_O_3_) [11,12,13,14,15], silicon carbide (SiC) [16,17,18], silicon nitride (Si_3_N_4_) [19,20,21], aluminum nitride (AlN) [22,23,24,25,26], boron nitride (BN) [27,28,29,30,31,32], carbon nanotubes (CNTs) [33,34,35] and grapheme [36,37,38] or other ceramic fillers. Boron nitride with a hexagonal lattice structure (h-BN) is a platelet-shaped ceramic and is often called “white graphite”. h-BN has the greatest potential application in polymer composites due to its superior intrinsic thermal conductivity (up to 400 W/m·k), excellent electrical insulation properties, thermal stability with low dielectric constant and low density, compared to those of Al_2_O_3_, SiC, Si_3_N_4_ and AlN. Therefore, in order to obtain the better thermal and mechanical properties of the composites, it is necessary to increase the interface between filler and epoxy matrix by surface modification with chemical treatment. The surface modification of inorganic filler improves the affinity of the filler matrix, which provides strong interconnection and homogeneous dispersion. Hence, the incorporation of functionalized ceramic materials in polymers enhances the demand of highly thermally properties for electronic encapsulation application.

In this study, a LC epoxy resin containing ketone mesogenic group was synthesized and a silane-coupling agent 3-aminopropyltriethoxysilane was used to modify the surface of h-BN fillers. The h-BN surface treatment was characterized, which confirms the silane functional groups were grafted onto the h-BN surface. The surface modification of BN fillers dispersed in LC epoxy resin matrix and then cured using aromatic amines as curing agent to produce high-performance composites. Its morphology and the dispersion states were examined. The thermomechanical properties, thermal stability and thermal conductivity of the cured composites were also investigated.

## 2. Experimental Section

### 2.1. Materials

Hexagonal boron nitride (h-BN) with an average diameter of 1 μm was purchased from Aldrich. 3-aminopropyltriethoxysilane (APTES) was purchased from Acros (Geel, Belgium). 4,4′-diaminodiphenylsulfone (DDS) was purchased from Tokyo Chemical Industry (TCI, Tokyo, Japan). Sulfuric acid, nitric acid and hydrochloric acid were purchased from TEDIA (Fairfield, OH, USA). Ethanol was purchased from ECHO (Lake Zurich, IL, USA). All chemicals and solvents were analytical grade and used directly without further purification as received.

### 2.2. Synthesis of 1,5-bis(ρ-glycidyloxy-phenyl)-1,4-pentadiene-3-one (K–LCE)

The K–LCE was synthesized following the method described in our previous work [9]. The synthetic route of K–LCE is shown in Scheme 1. The product as a yellow powder was recrystallized from ethanol.

K–LCE: calcd. 378; found: 379. ^1^H NMR (acetone–D_6_): δ = 2.75, 2.90 (4H, epoxy–CH_2_), 3.34 (2H, epoxy–CH), 3.93, 4.27 (4H, glycidyl–CH_2_), 6.92, 7.53 (8H, aromatic), 7.09 (2H, Ph–CH=), 7.69 (2H, =CH-). The epoxy equivalent weight (EEW) was determined using the HClO_4_/potentiometric titration method and found to be 196.2 g·eq^−1^.

### 2.3. Surface Modification of BN Fillers

In order to decrease the agglomeration of h-BN fillers, sulfuric acid, nitric acid and silane were used for surface treatment, and the surface of h-BN was functionalized to increase the degree of dispersion in K–LCE. First, 500 mg of BN powder was added into a 400-mL mixture solution of sulfuric acid and nitric acid (volume ration = 3:1), and the mixture was ultrasonicated for 10 h. The mixture was stirred with reflux condensation at 80 °C for 72 h and then cooled to room temperature. The resulting solution was filtered off, washed with deionized water and then dried at 60 °C under vacuum for 24 h. This pretreatment method was first, carried out to introduce the hydroxyl groups onto the surface of h-BN fillers through strong oxidation, named OH–BN.

Second, OH–BN fillers were modified with a silane-coupling agent APTES for grafting of hyperbranched. In a 250-mL three-neck flask, the mixture of deionized water and APTES (mole ratio = 12:1) were magnetic stirred for 30 min. The hydrolytic solution was added into ethanol and adjusted the pH to 3 with hydrochloric acid. Then the 5 g OH–BN filler was added into the mixed solution. The resulting mixture was heated to 70 °C and magnetic stirred for 5 h. After cooling, the silane modified BN filler was filtered, washed with ethanol and then obtained after dried at 60 °C under vacuum for 24 h, name Si–BN [29].

### 2.4. Preparation of K–LCE/BN Composites

The K–LCE-based composites were prepared as follows: First of all, a stoichiometric ratio of the K–LCE and curing agent DDS were mixed and the weight ratio of the Si–BN which set at 0.50, 1.00, 2.00 and 5.00 wt% were added to the mixture and pestle together in an agate mortar, respectively. Second, the mixture powders were poured into die, adequately degassed using manual hydraulic press and then placed into an aluminum mold. Finally, the K–LCE/BN composites were obtained after pre-cured at 120 °C for 1 h, 170 °C for 2 h and post-cured at 210 °C for 3 h. Figure 1 illustrates the process of the K–LCE/BN composites fabrication.

### 2.5. Characterization

FTIR spectra were recorded on PerkinElmer GX50003 instrument ranging from 4000 to 400 cm^−1^. The spectra of specimens were obtained by dispersing them in potassium bromide (KBr) and scanning. The surface morphologies of the K–LCE/BN composites were examined by scanning electron microscopy (SEM; Phenom XL, Waltham, MA, USA). The liquid crystalline property of the K–LCE/BN composites was observed by polarized optical microscopy (POM, Olympus BS51, Tokyo, Japan) with a heating stage during the curing process. The storage modulus and glass transition temperature (T_g_) were measured using a dynamic mechanical thermal analyzer (DMA; DMA Q800, TA Instruments, New Castle, DE, USA) with three-point bending mode at a frequency of 1 Hz. Specimens with dimensions of 20 mm × 15 mm × 2 mm were measured from the temperature range of 0 to 270 °C at a heating rate of 5 °C/min. The in-plane CTE was determined using a thermal mechanical analyzer (TMA; TMA Q400, TA Instruments, New Castle, DE, USA). The CTEs were tested by the expansion mode under a 0.05-N tension force from the temperature range of 0 to 270 °C at a heating rate of 5 °C/min. The impact testing for V-shape notched specimens was performed with a Charpy impact tester according to ASTM D6110. The impact strength was measured at room temperature and the impact velocity was 3.1 m/s. The decomposition temperature and weight loss were measured using a thermogravimetric analyzer (TGA; TGA Q500, TA Instruments, New Castle, USA) from room temperature to 800 °C at a heating rate of 10 °C/min under nitrogen gas. The thermal conductivities of the composite at room temperature were determined using a hot disk thermal conductivity analyzer (TPS2500, Gothenburg, Sweden) equipped with a Kapton disk-shaped sensor.

## 3. Results and Discussion

### 3.1. Characterization of BN Surface Modification

FTIR was performed to validate the grafting of the silane-coupling agent APTES on the h-BN surface during acid treatment and silanization. The FTIR spectra of the acid-treated OH–BN (a), APTES (b) and the silane-modified Si–BN (c) are shown in Figure 2 for comparison. The broad absorption peaks of OH–BN and APTES showed at around 3430 cm^-1^, which corresponded to the hydroxyl groups and amino groups, respectively. The strong absorption peaks at 1398 and 780 cm^-1^ are attributed to the vibration of B–N in-plane stretching and B–N–B out-of-plane bending, respectively. After OH–BN modified with silane, the new peaks at 3013, 2824, 1140 and 1036 cm^-1^ appeared in the IR spectrum of Si–BN, which corresponded to symmetric and asymmetric stretching vibrations of the methylene group of APTES and O–Si–O stretching, respectively. From the chemical structure characterization results that the silane-coupling agent APTES is successfully grafted on the h-BN surface.

### 3.2. Morphology Observation of K–LCE/BN Composites

The morphology of K–LCE/BN composites and the dispersion states of h-BN fillers in the K–LCE matrix were examined. Representative SEM images of the fractured surfaces of the K–LCE and K–LCE with different h-BN filler contents are shown in Figure 3. The neat K–LCE shows a smooth fractured surface in Figure 3a. The fractured shape of crack propagation reveals as river patterns, which indicated a brittle structure. For the composites with 2.00 wt% filer content, Figure 3d shows that h-BN fillers were well dispersed uniformly in the K–LCE matrix compared to those of the composites with 0.50 and 1.00 wt% filler content, as shown in Figure 3b,c. The surface morphologies of the Si–BN and raw h-BN-based composites with 5.00 wt% filler content are shown in Figure 3e,f, respectively. By comparisons, the raw h-BN-based composites showed much voids and the agglomeration of fillers. On the other hand, the surface-modified BN revealed good homogeneity and dispersion in K–LCE matrix without voids and agglomeration. This is due to the bonding between silane-coupling agent and the h-BN surface is to decrease the interaction of filler–filler and to enhance interfacial properties of the surface-modified BN fillers in the K–LCE matrix, such as good compatibility and uniform dispersion [27]. In addition, all the composites showed a rough fractured surface because of the K–LCE matrix shear yielding, which resulted by h-BN fillers added.

To further examine the texture formation during the curing process, the liquid crystalline phase of K–LCE/BN/DDS mixtures was observed using POM with a heating stage at the isothermal curing temperature. Figure 4 shows the POM image of cured K–LCE/BN composites. The cured K–LCE/BN composites showed the smectic phase which can also offer excellent physical properties. This is due to the mesogenic characteristic of liquid crystalline being introduced to produce a high orderly arrangement networks.

### 3.3. Thermal and Mechanical Properties of K–LCE/BN Composites

DMA can be used to investigate the viscoelastic properties and damping characteristics such as the Young’s modulus (E) and the tan δ, as shown in Figure 5. The storage modulus (E’) and loss modulus (E’’) were calculated from the applied stress and strain as a function of temperature or frequency. Furthermore, the T_g_ was calculated from the maximum peak position of the tan δ curve measurement. These properties reflect the molecular relaxation of the individual materials and reveal interactions among each material of the polymer composites. Figure 5a shows the storage modulus of the neat K–LCE and K–LCE/BN composites at a heating rate of 5 °C/min; the tan delta curves of the neat K–LCE and its composites filled with different h-BN filler content are shown in Figure 5b. The values of the pertinent parameters are summarized in Table 1. The storage modulus and T_g_ of the K–LCE /BN composites (above 2460 MPa, 245 °C) were higher than that of neat K–LCE (1968 MPa, 228 °C), which indicates that the mechanical properties enhancement can be attributed to the formation of covalent bridge bonding between the LC resin and functionalized h-BN fillers. Thus, silane-coupling agent played an important role in interaction and compatibility between filler and epoxy resin matrix. Nevertheless, the storage modulus and T_g_ of the 5.00 wt% modified BN-filled composites did not increased significantly compared to those of the loading of 2.00 wt% modified BN fillers. This result is due to the presence of some aggregations on the vitrification stage of curing reaction cause the phenomenon.

TMA is a useful technique to measure the dimensional changes of a material as a function of temperature under non-oscillating stress. TMA also is a very sensitive tool which can calculate the T_g_ and the CTEs in both the glassy and rubbery regions. TMA thermographs of the neat K–LCE and K–LCE/BN composites are shown in Figure 6. The CTE values of the composites were gradually decreased with increasing h-BN fillers, as listed in Table 1. This is ascribed to the surface modification with silane-coupling agent APTES, which can improve the dispersion of Si–BN in the K–LCE matrix and enhance the crosslinking density of the composites, resulting in the reduction of thermal expansion.

The impact tester was used to study the mechanical properties of composites. The composites toughness in terms of the impact strength in the unit of kJ/m^2^ was determined by the absorbed energy while breaking a sample under an impact load. Figure 7 shows the impact strength of the neat K–LCE and K–LCE/BN composites. It can be seen that the impact strength slightly increased with increasing Si–BN content compared to the neat K–LCE. This indicates that the incorporation of modified BN fillers did not affect the mechanical properties of the epoxy molding compounds.

TGA was performed to further investigate the thermal stability of composites such as weight loss, decomposition temperature and char yields. The thermal stability of the neat K–LCE and its composites filled with different h-BN filler content were determined by TGA at a heating rate of 10 °C/min under nitrogen gas, as shown in Figure 8. The neat K–LCE and all composites exhibited similar decomposition temperature at 5% and 10% weight loss, respectively. This is due to the existence of rigid rod group of LC epoxy, which can resist to thermal attack. Notice that the decomposition temperature of 5.00 wt% modified BN-filled composites at 10% weight loss was slightly lower compared to that of 2.00 wt% modified BN-filled composites. This result is may attributed to the existence of some aggregations effect the crosslinking density of composites. In addition, the char yields of the composites at 800 °C were gradually increased with increasing Si–BN fillers, as listed in Table 2. This result can be explained that the h-BN fillers as mass transfer barriers can form isolation layers [30]. Based on the above results, the thermomechanical properties of the composites are indeed enhanced by the incorporation of modified BN fillers.

### 3.4. Thermal Conductivity of K–LCE/BN Composites

The thermal conductivities of the neat K–LCE and K–LCE/BN composites increased along with the BN filler contents, as shown in Figure 9. The thermal conductivity values of K–LCE/BN composites with 0%, 0.05%, 1.00%, 2.00%, and 5.00 wt% BN filler contents were 0.34, 0.39, 0.45, 0.53 and 0.58 W/m·K, respectively. With incorporation of Si–BN to neat K–LCE, the thermal conductivity increased by 70.58% (from 0.34 to 0.58) at 5.00 wt% Si–BN fillers content. The enhancement in the thermal conductivity can be explained that the silane-coupling agent APTES modification improved the compatibility of h-BN fillers, which can provide the better dispersion state of Si–BN in the K–LCE matrix, thus obtaining the good path of heat dissipation.

In addition, the thermal and mechanical properties of the cured conventional epoxy resin (Diglycidyl ether of bisphenol A, DGEBA) composites were also investigated. The K–LCE/BN composites exhibited superior thermal and mechanical properties than DGEBA/BN composites, as summarized in Table 3. The properties of composites can be greatly enhanced when the rigid rod mesogenic structures with LC behavior are introduced, which can form a highly crosslinked composites with orderly arrangement structure.

## 4. Conclusions

K–LCE/BN composites were successfully prepared from functionalized h-BN fillers embedded in an LC epoxy resin. The silane-coupling agent APTES was used to modify raw h-BN and acted as a bridge between h-BN fillers and K–LCE matrix. The FTIR spectrum of modified h-BN confirms that APTES is successfully grafted on the surface of h-BN fillers. The thermomechanical properties of the composites are indeed enhanced by the incorporation of modified BN fillers, and the LC epoxy/BN composites shows excellent physical properties compared to those of the traditional epoxy/BN composites. Therefore, The K–LCE/BN composite have the potential for use as a high-performance epoxy molding compounds in electronics encapsulation industry with a low CTE, high storage modulus and thermal conductivity.

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
