# Peer review of "Preparation and Thermomechanical Properties of Ketone Mesogenic Liquid Crystalline Epoxy Resin Composites with Functionalized Boron Nitride"

_polymers, 2020, doi:10.3390/polym12091913_

Round 1
Reviewer 1 Report
The manuscript focuses on thermomechanical behaviors of hexagonal boron nitride fillers incorporated into a ketone mesogenic liquid crystalline epoxy matrix. The prepared K-LCE/BN composites show higher thermal stability, char yield, thermal decomposition temperature, and thermal conductivity. The developed composites are interesting and would be useful for electronic materials. I have a few questions. The author said that the composites show the smectic phase after curing, as shown in Fig.4. Please explain why you can determine the smectic phase from the POM image. In addition, I would like to know why the smectic phase enhances the thermo- and mechanical-stability. If it is nematic phase, these enhancements were not obtained, were there?
Author Response
We determined the type of liquid crystalline phase according to the previous literature (1. Our previous paper, Y. S. Lin, S. L. C. Hsu, 2017, “Synthesis, characterization, and thermomechanical properties of liquid crystalline epoxy resin containing ketone mesogen”, Polym. Eng. Sci., vol.57, pp. 424-431. 2. D. Ribera, A. Mantecon, A. Serra, 2001, “Synthesis and Crosslinking of a Series of Dimeric Liquid Crystalline Epoxy Resins Containing Imine Mesogens”, Macromol. Chem. Phys., vol.202, pp.1658–1671.). In the literature, they showed the similar optical micrographs of smectic liquid crystal.
The properties of epoxides can be greatly enhanced if rigid rod structures are introduced into their backbone during synthesis, no matter they are smectic or nematic liquid crystal. Most rigid rod epoxies exhibit liquid crystalline behavior and networks with an orderly arrangement of molecules.

Reviewer 2 Report
In this paper, the hexagonal boron nitride fillers were incorporated in the ketone mesogenic liquid crystalline epoxy matrix to prepare a high-performance epoxy composite. The thermomechanical properties and morphologies of K-LCE/BN composites loading with different contents of modified BN fillers were investigated in detail. However, after reviewing this manuscript carefully, I think it lacks the impact needed for Polymers and therefore it should be rejected from Polymers.
- This study aims to prepare a high-performance epoxy composite for use in ICs, but the materials prepared in this research only have a little improvement in thermomechanical properties compared with the LC epoxy resin reported before by the author (See table 3). And the maximum thermal conductivity is 0.58W/m·K still does not meet the requirements of ICs for thermal conductivity mentioned in 43th-45th line.
- There is no obvious originality and novelty in the methods used in this study.
Author Response
1. In general, the ICs encapsulation material comprises an epoxy resin (10~30 %), a curing agent (5~15 %), and fillers (60~85 %). In this work, our filler (BN) content is less than 6 %. If the BN is further increased to a higher level, the thermal conductivity could meet the requirements of ICs encapsulation material.
2. In this study, we have synthesized a novel liquid crystalline epoxy resin, which can enhance the thermomechanical properties of epoxy system. In addition, with the addition of functionalized boron nitride filler, we can further improve the thermal conductivity of the epoxy for use in ICs encapsulation.

Reviewer 3 Report
this work utilized innovative route to enhance the thermomechanical behaviors of the epoxy molding compounds. the research significance is valuable, the results are scientific and integrity. I recommend this paper to be published. I suggest some experiment measurement can be added, the fracture strength is a important mechanical property of the epoxy resin typed material, and is important for the material in structural application. if such data are added, the research work will be more significant.
Author Response
We have added the impact strength data in the manuscript. The impact testing for V-shape notched specimens was performed with a Charpy impact tester according to ASTM D6110. The impact strength was measured at room temperature and the impact velocity was 3.1 m/s. The composites toughness in terms of the impact strength in the unit of kJ/m2 was determined by the absorbed energy while breaking a sample under an impact load. The impact strength of the neat K-LCE and K-LCE/BN composites are shown in following figure. It can be seen that the impact strength slightly increased with increasing Si-BN content compared to the neat K-LCE. This indicates that the incorporation of modified BN fillers didn’t affect the mechanical properties of the epoxy molding compounds.

Round 2
Reviewer 2 Report
The authors have successfully addressed my previous concerns. The paper is recommended for publication.